# Berberine Phospholipid Is an Effective Insulin Sensitizer and Improves Metabolic and Hormonal Disorders in Women with Polycystic Ovary Syndrome: A One-Group Pretest–Post-Test Explanatory Study

**DOI:** 10.3390/nu13103665

**Published:** 2021-10-19

**Authors:** Mariangela Rondanelli, Antonella Riva, Giovanna Petrangolini, Pietro Allegrini, Attilio Giacosa, Teresa Fazia, Luisa Bernardinelli, Clara Gasparri, Gabriella Peroni, Simone Perna

**Affiliations:** 1IRCCS Mondino Foundation, 27100 Pavia, Italy; mariangela.rondanelli@unipv.it; 2Department of Public Health, Experimental and Forensic Medicine, University of Pavia, 27100 Pavia, Italy; 3Research and Development Department, Indena SpA, 20139 Milan, Italy; antonella.riva@indena.com (A.R.); giovanna.petrangolini@indena.com (G.P.); pietro.allegrini@indena.com (P.A.); 4Department of Gastroenterology and Clinical Nutrition, Policlinico di Monza, Via Amati 111, 20900 Monza, Italy; attilio.giacosa@gmail.com; 5Department of Brain and Behavioral Science, University of Pavia, 27100 Pavia, Italy; teresa.fazia01@ateneopv.it (T.F.); luisa.bernardinelli@unipv.it (L.B.); 6Endocrinology and Nutrition Unit, Azienda di Servizi Alla Persona ‘‘Istituto Santa Margherita’’, University of Pavia, 27100 Pavia, Italy; clara.gasparri01@universitadipavia.it; 7Department of Biology, College of Science, Sakhir Campus, University of Bahrain, Zallaq 32038, Bahrain; simoneperna@hotmail.it

**Keywords:** polycystic ovary syndrome, inflammation, visceral adipose tissue, homeostasis model assessment, berberine, phytosome, acne symptoms

## Abstract

Polycystic Ovary Syndrome (PCOS) is the most frequent endocrine disease in females of reproductive age and is characterized by multifactorial unhealthy conditions related to hormonal unbalance and also to dysmetabolism and inflammation. Recently, increasing evidence has shown that natural plant-based products may play a role in PCOS management. The aim of this one-group pretest–post-test explanatory study was to evaluate, in normal–overweight PCOS women with normal menses, the effectiveness of berberine on: Insulin resistance (IR) by Homeostasis Model Assessment (HOMA); Inflammation by C-Reactive Protein (CRP), Tumor Necrosis Factor α (TNF-α); Lipid metabolism; Sex hormone profile and symptoms correlated to hyperandrogenism, such as acne, by Global Acne Grading System (GAGS) and Cardiff Acne Disability Index (CADI); Body composition by DXA. Finally, adverse effects were assessed by liver and kidney functions and creatine phosphokinase (CPK). All these parameters were collected at baseline and 60 days after supplementation with a new bioavailable and safe berberine formulation. Twelve females (aged 26.6 ± 4.9, BMI 25.3 ± 3.6) were supplied for 60 days with two tablets/day (550 mg/table) of the bioavailable berberine. Results showed a statistically significant decrease in HOMA, CRP, TNF-α, Triglycerides, testosterone, Body Mass Index (BMI), Visceral Adipose Tissue (VAT), fat mass, GAGS and CADI scores, and a statistically significant increase in sex hormone-binding globulin (SHBG). Liver and kidney functions and CPK are not statistically significantly different. Therefore, berberine can represent a safe novel dietary supplement, helpful in treatment strategy for PCOS.

## 1. Introduction

Polycystic Ovary Syndrome (PCOS) is the most frequent endocrine disease in females of reproductive age, and is characterized by multifactorial unhealthy conditions related to hormonal unbalance, and also to dysmetabolism and inflammation. As a matter of fact, about 5–10% of premenopausal women are affected by Polycystic Ovary Syndrome (PCOS) [1,2,3,4]. The most typical aspects of PCOS related to a cystic or “micropolycystic” ovary were first reported by Antonio Vallisneri in 1721 [5]. Subsequently, this syndrome has been described analytically by Stein and Leventhal in 1935 [6].

The symptoms of PCOS are variable and heterogeneous [7,8]. The most common clinical features are chronic anovulation (80%), irregular menses (80%), and hyperandrogenism that may be associated with hirsutism (60%), acne (30%), seborrhea, and obesity (40%) [5].

From a pathogenic point of view, hyperinsulinemia and insulin resistance are basic factors that favor the development of hyperandrogenism and chronic anovularity by means of LH increase [9], a reduction in sex hormone-binding globulins (SHBGs), an increase in unbound circulating androgens [10], a reduction in androgen clearance and aromatase activity, and increased steroidogenesis in the adrenal glands and ovarian theca [11,12,13,14,15,16].

The treatment of PCOS is often a long-term therapy, and the most commonly used drugs are combined oral contraceptives (COCs), antiandrogenic progestins, and insulin-sensitizing drugs [17].

These pharmacological treatments could be associated with undesirable side effects and could particularly interfere with a hypothetical pregnancy [18,19].

For these reasons, botanicals have been often thought to provide a helpful supportive treatment for patients with PCOS. In particular, berberine is an alkaloid widely used in Chinese herbal medicine against infections, hypercholesterolemia, diabetes type 2 and cancer [20].

Berberine is also shown to be effective against insulin resistance and obesity, particularly against visceral adipose tissue in in vitro and in murine models [2,21,22].

All of the above preclinical data have been confirmed in human studies, where berberine can modulate the diversity of gut microbes at a dose of 500 mg/day. In addition, Berberine is found to have a beneficial impact on gene regulation for the absorption of cholesterol at a daily dose of 300 mg in humans, an amelioration on glucose accumulation at 1.0 g daily dose was also observed, as reported in a recent systematic review [23].

Berberine is a very promising botanical compound because its target is an AMP-activated protein kinase (AMPK) common to fatty acid oxidation, glucose generation, and insulin resistance [24]. Due to this mechanism of action, berberine has been tested in the clinical management of dyslipidemia, diabetes type 2, and obesity [24,25,26,27]. Thanks to minor side effects on long-term treatment, berberine is the only botanical compound included in European guidelines for the management of dyslipidemia, and it is also used in patients who cannot tolerate statins [28,29,30].

Moreover, it has been demonstrated that berberine can enhance the expression of antioxidant enzyme activity by activating different pathways; these antioxidants can protect the biological membrane of the body from the damage of free radicals and reduce the consumption of glutathione and the generation of reactive oxygen species, thus inhibiting lipid peroxidation [31]. Berberine, if associated with a healthy lifestyle, improves women’s body composition and causes androgen’s reduction, as pointed out by Saleem et al. [22].

PCOS is an endocrine–metabolic disorder very similar to the metabolic syndrome, and they indeed share a common pathogenic factor: insulin resistance. Ong et al. [20] explained that insulin resistance is the key factor that could cause obesity and anovulatory cycles and that, actually, this should be the target of therapeutic molecules against PCOS and against metabolic syndrome [2,20]. These researchers showed that berberine induces an amelioration of insulin resistance and an improvement of the regularity of menstrual cycles when orally administered at a dose of 500 mg twice-per-day for 6 months [20].

A recent narrative review that included five eligible studies with an evaluation of 1078 female patients with PCOS [32] demonstrated that berberine appeared to be safe, and two studies found that berberine induces a redistribution of adipose tissue, reducing VAT in the absence of weight loss and improves insulin sensitivity, similar to metformin. One study demonstrated that berberine improves the lipid pattern too [33].

Moreover, studies have demonstrated that berberine counteracts insulin resistance in the cells, with an improvement of the ovulation rate per cycle, a promising activity for the effectiveness on fertility and live birth rates [34,35]. Therefore, berberine appears to be safe and potentially efficacious in premenopausal women with PCOS who want to get pregnant.

In conclusion, berberine has been defined by a recent review [36] as a multi-target, multi-path natural product that can interfere with the pathological process of PCOS from many aspects.

However, to date, there has been no human study that considers all the activities carried out by berberine on the clinical features of PCOS as a whole, as the published studies take into account only the pharmacological activities of berberine separately. Moreover, berberine is often used in combination with metformin, cyproterone acetate (CPA), and other drugs in order to achieve a better therapeutic effect on PCOS, and therefore there are few studies that evaluate the activity of berberine on its own.

Given this background, the aim of the present study was to evaluate the effectiveness of a new berberine formulation on insulin resistance as a primary endpoint, assessed by Homeostasis Model Assessment (HOMA), and, as secondary endpoints, on inflammation, by C-Reactive Protein (CRP), Tumor Necrosis Factor α (TNF-α) and visceral adipose tissue (VAT) (evaluated by dual-energy X-ray absorptiometry (DXA)), on lipid metabolism, by total cholesterol, HDL cholesterol, LDL cholesterol, Triglycerides, on sex hormone profile, by free and total testosterone, sex hormone-binding globulin, free androgen index (FAI), on symptoms correlated to hyperandrogenism, such as acne, by Global Acne Grading System (GAGS) and Cardiff Acne Disability Index (CADI), and on body composition by DXA. Finally, adverse effects were assessed by liver and kidney functions and creatine phosphokinase (CPK).

## 2. Materials and Methods

### 2.1. Study Endpoints

We considered, as primary endpoint, the assessment of HOMA, and, as secondary endpoints, the evaluation of: inflammation by CRP, TNF-α, and VAT (evaluated by DXA); lipid metabolism by total cholesterol, HDL cholesterol, LDL cholesterol, triglycerides; sex hormone profile by free and total testosterone, sex hormone-binding globulin, FAI, and symptoms correlated to hyperandrogenism, such as acne, by GAGS and CADI; anthropometric parameters by Body Mass Index (BMI), waist circumference, hip circumference and body composition by DXA which evaluates fat mass, free fat mass (FFM) and VAT. Finally, adverse effects were assessed by aspartate transaminase (AST), alanine transaminase (ALT), gamma-glutamyl transferase (GGT)), creatinine, and CPK.

All parameters were collected at the start and at the end of the supplementation after 60 days.

HOMA, anthropometric parameters, and body composition by DXA were assessed also after 30 days.

### 2.2. Study Design

This is a one-group pretest–post-test explanatory study in which all the participants received the supplement and were observed over time. There was no control group and, as a consequence, the study is not randomized.

### 2.3. Population

The study was conducted in normal and overweight women (BMI 20–30 kg/m2) with newly detected PCOS as defined by the Rotterdam ESHRE/ASRM-Sponsored PCOS Consensus Workshop Group [4], with regular menses, consequently admitted as outpatients, to the Dietetic and Metabolic Unit of the “Santa Margherita” Institute, University of Pavia, Italy. These adult females (aged between 20 and 35 years) were included in this pilot study between September 2020 and January 2021. The subjects were not taking any medication and were free of overt liver, renal, and thyroid disease. Subjects who smoked, or who drank more than two standard alcoholic beverages/day (20 g of alcohol/day), were excluded from the study. Physical activity was recorded. Sedentary subjects were admitted to the study. The experimental protocol was approved by the Ethics Committee of the University of Pavia (ethical code number: 9321/14122019) and registered at Clinicaltrials.gov (NCT04932070). All the volunteers gave their written informed consent.

### 2.4. Dietary Supplement

The recommended dietary treatment was associated with 2 daily oral doses (one before lunch and one dinner) of 550 mg of berberine tablets. The supplementation period was 60 days. The tablets were delivered at the time of the first blood sample.

Berberine Phytosome^®^ (BBR-PP, berberine phospholipids/PRO, Berbevis^®^, Patent Application WO2019/150225) is a solid dispersion containing berberine extract in a rational combination with sunflower lecithin, pea protein (Nutralys^®^ S85F, supplied by Roquette Freres, Lestrem, France), and grape seed extract (Enovita^®^), as previously described (Petrangolini et al., manuscript in submission 2021). BBR-PP is standardized to contain 28–34% of berberine (by HPLC).

Berberine Phytosome^®^ (BBR-PP) 550 mg/dose was formulated as film-coated tablets (kindly donated by Indena, Milan, Italy) containing the following food grade ingredients: Dicalcium phosphate dihydrate, Microcrystalline cellulose, Sodium croscarmellose, Silicon dioxide, Talc, Magnesium stearate, and Hydroxypropylmethylcellulose-based film-coating. Film-coated tablets were characterized for appearance, average mass, uniformity of mass, HPLC content of berberine, disintegration time, and microbiological quality.

Adherence to treatment was assessed by counting the number of supplements remaining when the participants returned to the laboratory. A value of 90% of the total tablets’ consumption of the supplementation was achieved.

### 2.5. Adverse Events

Adverse events were based on spontaneous reporting by subjects, as well as on open-ended inquiries by members of the research staff. Moreover, routine blood biochemistry parameters (creatinine, liver function) were evaluated at the start and at the end of supplementation.

### 2.6. Biochemical Parameters

In order to avoid venipuncture stress, blood samples were obtained through an indwelling catheter inserted in an antecubital vein. Blood samples were immediately centrifuged and stored at −80 °C until assayed. Fasting blood glucose (FBG), total cholesterol, low-density lipoprotein-cholesterol (LDL), high-density lipoprotein-cholesterol (HDL), and triglyceride levels were measured by automatic biochemical analyzer (Hitachi 747, Tokyo, Japan).

The serum insulin was evaluated by a double antibody RIA (Kabi Pharmacia Diagnostics AB, Uppsala, Sweden) and expressed as pmol/L. The intra- and inter-assay coefficients of variation were below 6%, and the low detection limit was 10.7 pmol/L. To determine insulin resistance, subjects were instructed to fast for 12 h before obtaining the blood sample. Furthermore, the subjects refrained from any form of physical exercise for 48 h before the blood sampling. Insulin resistance was evaluated using the HOMA [37].

CRP level was measured by particle-enhanced immunonephelometry on a Behring Nephelometer analyzer using the relevant kit (Dade Behring, Marburg, Germany).

Serum levels of TNF-α were assessed using the TNF-α Enzyme Linked Immunosorbent Assay (ELISA) test (Biosource International, Human TNF-α, Belgium).

Total Testosterone (TT) was evaluated as serum total concentrations using the electrochemiluminescence immunoassay (ECLIA) on a cobas 8000 modular analyzer (E602, Roche Diagnostics GmbH, Mannheim, Germany). Free testosterone (FT) was calculated as the product of total testosterone and free testosterone percentage. The free testosterone percentage was determined by equilibrium dialysis and was corrected for dilution using the formula of Vermeulen et al. [38].

Sex hormone-binding globulin (SHBG) levels were detected using a chemiluminescence assay (Roche Cobas E601, Basel, Switzerland).

The free androgen index was calculated as FAI = (TT/SHBG) × 100 [39].

CPK was studied with Roche diagnostic kits in a COBAS C-8000 Roche autoanalyzer, by an enzymatic UV method.

Finally, for the assessment of safety, routine blood biochemistry parameters of liver function were evaluated: alanine aminotransferase, aspartate aminotransferase, gamma glutamyl transferase, and total bilirubin were measured with enzymatic colorimetric methods.

### 2.7. Anthropometric Measurements and Dietary Counseling

Body weight and height were measured following a standardized technique [40] and the BMI was calculated (kg/m^2^). Anthropometric parameters were always collected by the same investigator.

Subjects were trained to maintain a prudent balance of macronutrients: 25–30% of energy from fat (cholesterol < 200 mg), 55–60% of energy from carbohydrates (10% from simple carbohydrates), with 25 g of bran and 15–20% of energy from protein. A registered dietician performed initial dietary counseling. A 3-day weighed-food record of 2 weekdays and 1 weekend day was performed during the first and the last week of the study. Dietary records were analyzed using a food-nutrient database (Rational Diet, Milan, Italy).

### 2.8. Body Composition

Body composition (FFM, fat mass (FM)) was measured by DXA with the use of a Lunar Prodigy DXA (GE Medical Systems). The in vivo coefficients of variation (CVs) were 0.89% and 0.48% for whole body fat (FM) and FFM, respectively.

Visceral adipose tissue volume was estimated using a constant correction factor (0.94 g/cm^3^). The software automatically places a quadrilateral box, which represents the android region, outlined by the iliac crest, and with a superior height equivalent to 20% of the distance from the top of the iliac crest to the base of the skull [41].

### 2.9. Assessment of Acne Status

GAGS was used to grade facial acne. This grading system calculates the severity of acne through the combined assessment of the types of acne lesions (comedones, papules, pustules, and nodules) and their anatomic location (forehead, cheeks, nose, and chin). The GAGS considers five locations on the face, with a factor at each location based roughly on surface area, distribution, and density of pilosebaceous units.

Each type of acne lesion is given a value depending on severity: no lesions = 0, comedones = 1, papules = 2, pustules = 3, and nodules = 4. Each of the location was graded separately on 0–4 scale, with the most severe lesion within that location determining the local score. The severity was then graded according to the global score, which is the summation of all local scores. A score of 1–6 was considered mild; 7–18, moderate; 19–26, severe; 27–32, very severe. The maximum score was 32 [42].

The CADI is a well-validated, self-reported questionnaire designed for measuring disability induced by acne in teenagers and young adults. The CADI consists of five questions with a Likert scale and four response categories (0–3). The five questions relate to feeling of aggression, frustration, interference with social life, avoidance of public changing facilities, and appearance of the skin—all over the last month—, as well as an indication of how bad the acne was at the time of competing the questionnaire. The CADI score was calculated by summing the score of each question resulting in a possible maximum of 15 and minimum of 0. CADI scores were graded as low (0–4), medium (5–9), and high (10–15). The lower the cumulative CADI score, the lower the level of disability experienced by the student, while a higher score indicated a higher level of disability. The internal consistency reliability of the CADI was found to be high, Cronbach’s α coefficient = 0.962 [43].

### 2.10. Statistical Analysis

Assuming that this was a pilot study, the sample size was defined by the feasibility of recruitment. For the recruited sample size, power analysis was determined post-hoc based on 100 simulations using SIMR package and was equal to 0.82, with an α equal to 0.05 [44,45].

Differences between baseline and final assessment in calorie and macronutrient intake were assessed by the 3-day weighed-food record of 2 weekdays, and 1 weekend day performed during the first and the last week of the study was investigated using *t*-test.

In order to evaluate statistically significant pre- and post-treatment changes, we fitted a linear mixed model (LMM) for each investigated endpoint, with time as fixed effect and a random intercept for each subject, in the form of 1 per subject, to account for the intra-subject correlation produced by the two different measurements carried out on the same patients [46]. All the models were adjusted for age. *p*-values < 0.05 on a 2-sided test are considered as statistically significant. Normality was assessed graphically and with a Shapiro–Wilk test. Benjamini–Hochberg correction, fixing the false discovery rate (FDR) at α < 0.05, was used to account for multiple comparison [47].

Descriptive statistics are reported as Mean ± Standard Deviation (SD). All the analysis was performed on R 3.5.1 Software using the NLME and stats packages [48].

## 3. Results

A total of 12 females with a mean (±SD) age of 26.67 (±4.92) years were included in the study.

The HOMA, as primary endpoint, was used to detect the insulin resistance and function of pancreatic beta-cells (which produce insulin). The secondary endpoints were used to detect the glycemic profile, and the insulin resistance were the determination of glycemia and insulin levels, respectively.

The following tables (Table 1, Table 2, Table 3, Table 4, Table 5 and Table 6) report the descriptive statistics measured for each investigated endpoint at baseline (t0), after 30 days (t1), and after 60 days (t2), together with the variation in percentage in respect to the baseline value (t0).

These results showed a statistically significant decrease in the primary endpoint HOMA (β = −0.69, *p* = 0.003), used to detect the insulin resistance and function of pancreatic beta-cells (which produce insulin), and the reduction of glycemia (β = −4.50, *p* = 0.0001) and insulin (β = −2.83, *p* = 0.005) levels after 30 and 60 days of supplementation (Table 1) (Figure 1).

The supplementation of BBR-PP modulated also the lipid profile of subjects with a significant decrease in: VLDL (β = −3.11, *p* = 0.03) and Triglycerides (β = −15.42, *p* = 0.03); and a slight modulation on the total cholesterol, LDL, and HDL, as reported in Table 2.

Regarding the hormonal pattern, BBR-PP was statistically effective on the modulation of testosterone (β = −0.15, *p* = 0.007) and SHBG (β = 9.04, *p* = 0.03) (Table 3).

Of particular interest, the acne symptoms were significantly decreased after supplementation with BBR-PP, according to the international validated score: GAGS (β = −8.17, *p* < 0.0001) and CADI (β = −8.158 *p* < 0.0001), as shown in Table 3 and Figure 2.

Moreover, after BBR-PP supplementation, we registered an amelioration of body composition: BMI (β = −0.41, *p* = 0.003), waist circumference (β = −1.42, *p* = 0.005), hip circumference (β = −0.96, *p* = 0.003), W/H ratio (β = −0.006, *p* = 0.04), VAT (−49.96, *p* = 0.003), total mass (β = −1.17, *p* = 0.002), fat mass (β = −857.25, *p* = 0.003), as summarized in Table 4.

Moreover, the statistical modulation on body composition was observed in VAT and fat mass, as shown in Figure 3.

Significant results were observed even in the inflammatory response after BBR-PP supplementation: CRP (β = −0.14, *p* = 0.02), TNF α (β = −6.17, *p* = 0.009), as shown in the following table (Table 5).

Finally, no adverse effects were reported by subjects or observed by liver and kidney functions and creatine phosphokinase (CPK) (Table 6).

Linear mixed models, adjusted for age, were fitted to evaluate significant pre- and post-treatment changes (time) on the analyzed primary and secondary outcomes (full results are reported in Table 7).

## 4. Discussion

Previous studies [31,33] and one meta-analysis [49] have indicated that berberine is an effective insulin sensitizer with comparable activity to metformin in women with PCOS [50,51,52]. Moreover, even in the studies that evaluate whether berberine taken in combination with drugs such as metformin, which is effective as an insulin sensitizer, the results are confirmed [53]. For the first time, our pilot human study considers all the activities carried out by berberine on the clinical features of PCOS as a whole, due to its the multifactorial unhealthy condition related to hormonal unbalance, which influences also metabolism and inflammation. Our results, which demonstrate the effectiveness of berberine in reducing HOMA values, are also in agreement with these previous studies and meta-analysis.

It is still not clear exactly how this supplementation affects metabolic pathways, but several mechanisms can be postulated. In a PCOS rat model, it has been demonstrated that berberine may relieve PCOS pathology and IR values by inhibiting cell apoptosis and the inflammatory response through regulating the expression levels of TLR4, LYN, PI3K, Akt, NF-kB, TNF-α, IL-1, IL-6, and caspase-3 [54] and by the PI3K/AKT pathway [55], therefore berberine exerts a protective effect on rats with PCOS through the inhibition of the inflammatory response and cell apoptosis.

Subjects with PCOS have also been found to be under a chronic low-grade inflammation status, including high levels of leukocytes and a disorder of the pro-inflammatory cytokines [56,57,58]. Only Cicero and colleagues have studied the effect of berberine on the CRP and on inflammation [33]. They found that berberine in obese PCOS women causes a reduction of CRP statistically relevant compared to obese subjects. Our study confirmed these previous results reported by Cicero, with a statistically significant reduction in CRP after 2 months of supplementation with berberine observed. Moreover, in our study, for the first time in the literature, two other important markers of inflammation were considered: the pro-inflammatory cytokine TNF-α and visceral adipose tissue [59]. Both markers decreased significantly after supplementation.

Finally, this is the first pilot study that provides early evidence that, in a group of PCOS women, eight weeks of BBR-PP supplementation may have positive effects on body composition. In fact, it has been found that berberine induced a redistribution of adipose tissue, by reducing visceral fat mass and fat mass, in the absence of a fat-free mass reduction, even though supplemented subjects did not follow a low calorie diet. The influence of any changes in the diet is to be excluded, as there were no statistically significant differences between baseline and the final assessment in calorie and macronutrient intake assessed by the die 3-day weighed-food record of 2 weekdays and 1 weekend day performed during the first and the last week of the study.

A recent review reported that berberine may improve IR and lipid metabolism by reducing lipid synthesis, promoting lipid consumption, and increasing fat factor, so as to regulate the endocrine system of PCOS patients [36]. Considering hormonal patterns, the results of this study demonstrated that testosterone and FAI decreased, whereas SHBG increased significantly after 2 months of treatment, as already demonstrated in previous studies [52,60,61].

The effectiveness of berberine in counteracting hyperandrogenism is very important in order to reduce the clinical symptoms related to the condition [7,8], such as hirsutism (60%), acne (30%), seborrhea, and obesity (40%) [5].

This study, for the first time in the literature, has shown that berberine supplementation could have a positive effect on acne, as demonstrated by the statistically significant decrease in the score of both GAGS and CADI in this specific population (normal or overweight PCOS women with normal menses).

Finally, a consideration regarding the dropouts of the study is also important, as one young woman was a dropout because she became pregnant. This data is interesting considering that there are studies shown that berberine supplementation, prior to in vitro fertilization treatments (IVF), improved the pregnancy outcome by normalizing the clinical, endocrine, and metabolic parameters in PCOS women [50,60].

No adverse events were observed, according to Cicero et al. [33] and Orio et al. [31], while other authors have reported gastrointestinal side effects [50]. The lack of side effects, in particular gastrointestinal discomforts, recorded in our study could be due to the particular formulation used.

The bioavailable BBR-PP allowed us also to use a lower dose of the alkaloid per unit (200 mg). The previous studies used a higher dosage than our study: from 300 mg × 3/day for 3 months [62] to 500 mg × 2/days [31,63] and 500 mg × 3/day [35,50,52].

The lack of side effects is very important considering that this supplementation should be chronic and that bowel discomfort is the main issue hampering the long-term use of berberine.

This study has some limitations. The first is the small sample size; given that this was a pilot study and that there was little prior evidence, the sample size was determined by the feasibility of recruitment. The second limitation regards the enrolled females, which included only normal or overweight PCOS women with normal menses and may limit generalization to the PCOS population. Therefore, all these findings must be interpreted with caution, and further studies are needed with a larger population size.

Finally, another limitation is that, given the study design, it cannot offer any clear insight into the possible mechanisms underlying the findings, which must, therefore, remain purely hypothetical.

Regarding the strengths of this study, the first is that berberine formulation used as a dietary supplement was highly standardized and supported with a good bioabsorption profile. The issues of standardization, characterization, preparation, absorption, and toxicity of botanicals is crucial for quality supplementation [64].

## 5. Conclusions

In conclusion, in this study it was shown that berberine may have a positive activity in reducing insulin resistance, acne, androgen, and inflammation, in regulating lipid metabolism, and in improving body composition, and therefore can represent a novel clinical supplementation strategy for PCOS, although the results are demonstrated only in a specific population (normal or overweight PCOS women with normal menses) and may limit generalization to the PCOS population. Therefore, all these findings must be interpreted with caution and further randomized clinical trials are needed with a larger population size.

## Figures and Tables

**Figure 1 nutrients-13-03665-f001:**
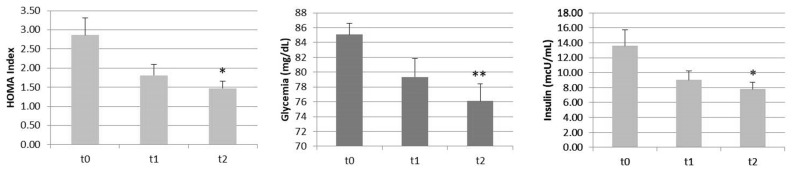
BBR-PP effectiveness on the primary and secondary endpoints on insulin resistance and the glycemic profile. * *p* ≤ 0.005; ** *p* < 0.0005.

**Figure 2 nutrients-13-03665-f002:**
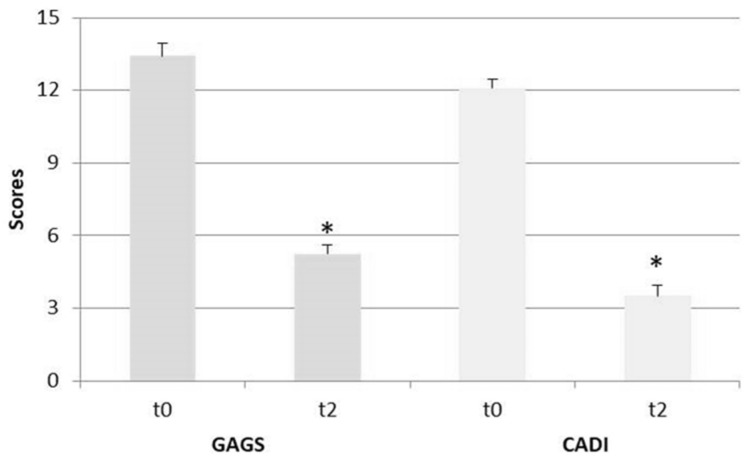
Berberine effectiveness on the amelioration of acne status. * *p* < 0.0001.

**Figure 3 nutrients-13-03665-f003:**
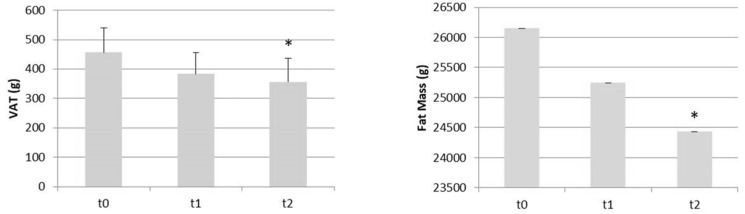
BBR-PP effectiveness on body composition, inducing a redistribution of adipose tissue with the reduction of the visceral fat tissue and fat mass. * *p* < 0.005.

**Table 1 nutrients-13-03665-t001:** Descriptive statistics for the primary endpoint and the glycemic and insulin profiles measured at baseline (t0), after 30 days (t1), and after 60 days (t2) of BBR-PP supplementation.

Parameters	t0	t1		t2	
	Value Mean (SD)	Value Mean (SD)	Variation vs. t0 (%)	Value Mean (SD)	Variation vs. t0 (%)
HOMA (pt)	2.86 (1.55)	1.81 (0.98)	−36.7	1.47(0.63)	−48.6 *
Glycemia (mg/dL)	85.08 (5.21)	79.33 (8.79)	−6.8	76.08 (8.13)	−10.6 **
Insulin (mcU/mL)	13.46 (7.25)	9.02 (4.31)	−33.0	7.79 (3.11)	−42.1 *

* *p* ≤ 0.005; ** *p* < 0.0005.

**Table 2 nutrients-13-03665-t002:** Descriptive statistics for the lipid profiles measured at baseline (t0) and after 60 days (t2) of BBR-PP supplementation.

Parameters	t0	t2	
	Value Mean (SD)	Value Mean (SD)	Variation vs. t0 (%)
Total Cholesterol (mg/dL)	160.08 (50.32)	155.08 (44.74)	−3.1
HDL (mg/dL)	55.17 (10.60)	57.75 (14.01)	4.7
LDL (mg/dL)	87.13 (42.25)	83.68 (34.31)	−4.0
VLDL (mg/dL)	17.78 (7.24)	14.67 (5.23)	−17.5 *
Triglycerides (mg/dL)	88.5 (36.23)	73.08 (26.12)	−17.4 *

LDL: low-density lipoprotein; HDL: high-density lipoprotein; VLDL: very-low-density lipoprotein. * *p* < 0.05.

**Table 3 nutrients-13-03665-t003:** Descriptive statistics for the hormonal pattern and acne status measured at baseline (t0) and after 60 days (t2) of BBR-PP supplementation.

Parameters	t0	t2	
	Value Mean (SD)	Value Mean (SD)	Variation vs. t0 (%)
SHBG (nmol/l)	62.73 (38.43)	71.78 (39.25)	+14.4 *
Free Testosterone (ng/mL)	0.46 (0.22)	0.31 (0.31)	−32.6 **
FAI (pt)	3.18 (3.16)	2.4 (1.70)	−24.5
GAGS	13.42 (1.83) (Moderate)	5.25 (1.29) (Mild)	−60.9 ***
CADI	12.08 (1.38) (High)	3.5 (1.62) (Low)	−71.0 ***

SHBG: Sex Hormone-Binding Globulin; CPK: Creatine Phosphokinase; FAI: Free Androgen Igndex; GAGS: Global Acne Grading System; CADI: Cardiff Acne Disability Index Score. * *p* < 0.05; ** *p* < 0.01; *** *p* < 0.0001.

**Table 4 nutrients-13-03665-t004:** Descriptive statistics for the body composition measured at baseline (t0) and after 60 days (t2) of BBR-PP supplementation.

Parameters	t0	t1		t2	
	Value Mean (SD)	Value Mean (SD)	Variation vs. t0 (%)	Value Mean (SD)	Variation vs. t0 (%)
BMI (Kg/m^2^)	25.39 (3.69)	24.70 (3.76)	−2.7	24.57 (3.74)	−3.2 **
VAT (g)	456.92 (287.32)	383,75 (254.10)	−16.0	357.00 (276.13)	−21.9 **
Waist circumference (cm)	87.46 (9.11)	86 (8.88)	−1.7	84.62 (8.27)	−3.2
Hip circumference (cm)	106 (12.60)	104,46 (12.65)	−1.5	104.08 (12.47)	−1.8 **
W/H Ratio	0.83 (0.06)	0.83 (0.06)	0.0	0.82 (0.06)	−1.2 *
Total mass (Kg)	68.32 (10.87)	66.71 (10.56)	−2.4	65.97 (65.97)	−3.4 **
Fat mass (g)	26147.58 (7230.92)	25243.17 (6769.28)	−3.5	24433.08 (6565.32)	−6.6 **
Lean mass (g)	39863.17 (6011.00)	39095.5 (5725.30)	−1.9	39249.67 (5571.37)	−1.5

W/H: Waist/Hip Ratio; BMI: Body Mass Index. * *p* < 0.05; ** *p* < 0.005.

**Table 5 nutrients-13-03665-t005:** Descriptive statistics for the inflammatory response measured at baseline (t0) and after 60 days (t2) of BBR-PP supplementation.

Parameters	t0	t2	
	Value Mean (SD)	Value Mean (SD)	Variation vs. t0 (%)
CRP (mg/dL)	0.33 (0.39)	0.19 (0.26)	−42.4 *
TNF α (pg/mL)	12.24 (6.20)	6.07 (2.70)	−50.4 **

CRP: C Reactive Protein; TNF α: Tumor Necrosis Factor. * *p* < 0.05; ** *p* < 0.001.

**Table 6 nutrients-13-03665-t006:** Descriptive statistics for each safety blood parameter measured at baseline (t0) and after 60 days (t2) of supplementation.

Parameters	t0	t2
	Value Mean (SD)	Value Mean (SD)
Total Bilirubin (mg/dL)	0.43 (0.21)	0.42 (0.17)
AST (IU/L)	20.08 (9.59)	18.08 (5.92)
ALT (IU/L)	20.08 (13.15)	17.42 (8.02)
GGT (U/L)	13.58 (4.85)	13.42 (4.23)
CPK (U/L)	96.5 (42.84)	67.5 (22.95)

AST: aspartate transaminase; ALT: alanine transaminase; GGT: gamma-glutamyl transferase.

**Table 7 nutrients-13-03665-t007:** Estimate (β), Standard Error, and *p*-value of the treatment effect on the primary and secondary endpoints, evaluated as difference between pre- and post-treatment, using a linear mixed model.

Endpoints	Time × Group β [95%CI]	*p*-Value Unadjusted	*p*-Value Adjusted
*Primary endpoint*			
HOMA	−0.69 [−1.06; −0.31]	0.0009	0.003
*Secondary endpoints*			
Total Cholesterol (mg/dL)	−5.00 [−18.21; 8.21]	0.42	0.49
HDL (mg/dL)	2.58 [−3.18; 8.35]	0.34	0.45
LDL (mg/dL)	−3.45 [−12.54; 5.64]	0.42	0.49
VLDL (mg/dL)	−3.11 [−5.55; −0.66]	0.02	0.03
CRP(mg/dL)	−0.14 [−0.23; −0.04]	0.008	0.02
TNF α (pg/mL)	−6.17 [−9.91; −2.44]	0.004	0.009
Triglycerides (mg/dL)	−15.42 [−27.65; −3.18]	0.02	0.03
Glycemia (mg/dL)	−4.50 [−6.20; −2.80]	<0.0001	0.0001
Insulin (mcU/mL)	−2.83 [−4.51; −1.15]	0.002	0.005
SHBG (nmol/L)	9.04 [2.07; 16.01]	0.02	0.03
CPK (U/L)	−29.00 [−50.58; −7.42]	0.01	0.02
Free Testosterone (ng/mL)	−0.15 [−0.24; −0.06]	0.003	0.007
FAI	−0.78 [−2.04; 0.47]	0.20	0.28
Total Bilirubin (mg/dL)	−0.01 [−0.13; 0.12]	0.91	0.91
AST (IU/L)	−2.00 [−7.85; 3.85]	0.47	0.52
ALT (IU/L)	−2.67 [−8.86; 3.52]	0.36	0.45
GGT (U/L)	−0.17 [−1.62; 1.29]	0.81	0.84
BMI (kg/m^2^)	−0.41 [−0.64; −0.19]	0.001	0.003
Waist Circumference (cm)	−1.42 [−2.27; −0.57]	0.002	0.005
Hip Circumference (cm)	−0.96 [−1.49; −0.42]	0.001	0.003
W/H ratio (pt)	−0.006 [−0.01; −0.0004]	0.03	0.04
VAT (g)	−49.96 [−75.85; −24.06]	0.0006	0.003
Total Mass (kg)	−1.17 [−1.73; −0.61]	0.0003	0.002
Fat Mass (g)	−857.25 [−1334.43; −380.06]	0.001	0.003
Lean Mass (g)	−306.75 [−661.89; 48.39]	0.09	0.13
GAGS (pt)	−8.17 [−9.28; −7.05]	<0.0001	<0.0001
CADI (pt)	−8.58 [−9.65; −7.52]	<0.0001	<0.0001

## Data Availability

The data presented in this study are available inside the article.

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
