# Peer review of "Berberine Phospholipid Is an Effective Insulin Sensitizer and Improves Metabolic and Hormonal Disorders in Women with Polycystic Ovary Syndrome: A One-Group Pretest–Post-Test Explanatory Study"

_nutrients, 2021, doi:10.3390/nu13103665_

Round 1

Reviewer 1 Report

It is necessary to include the placebo group. In order to prove the effect of the tested supplement, it is necessary to exclude the influence of changes in diet and weight reduction. Currently, PCOS is considered to be one of the major hormonal complications of obesity. Treatment of obesity is the cornerstone of PCOS therapy in overweight and obese women. It has been proven in many studies that such conduct causes changes in metabolic and hormonal parameters. Therefore, the work only shows the known facts, because the lack of a placebo group makes it impossible to demonstrate the effect of the supplement used.

Reviewer 2 Report

In spite of the study flaw that you honestly mentioned related to lack of randomization (as well as placebo control) and very small sample size, your article has merit in its thoroughness of the parameters measured within the study participants.

However, I recommend a further amplification of the introduction to the importance of berberine as a potential phytotherapeutic compound.

Additionally, a revision of the English syntax as well as the elimination of some redundant words is also necessary.

Round 2

Reviewer 1 Report

Authors should wait with the publication of the study until they increase the study group and add the placebo group.

Author Response

This is a case Reports and Series trial.
It is an explanatory pilot study, without a control group, which in any case, with all its limitations, remains a possible design.

The results must obviously be read and interpreted taking this into account (as written in the limitations). However, the results are interesting both for the investigated parameters and for the results obtained that demonstrate the effectiveness of the product. With a one group study it is possible to make an inference even if limited to a population that has the same characteristics as ours in the study. Moreover, we have been very careful that we cannot speak of causality.

On the other hand, adding the placebo group at this point would still not be good because it would not be randomized. So it would in any case remain a non-RCT study.

Extensive editing of English language and style were done. We enclose revision certification for the English language
